# MAKING TEXT EMBEDDERS FEW-SHOT LEARNERS

**Chaofan Li**[1,2]*, **Minghao Qin**[2,3]*, **Shitao Xiao**[2]*, **Jianlyu Chen**[2,4], **Kun Luo**[2,3], **Defu Lian**[4†],
**Yingxia Shao**[1†], **Zheng Liu**[2†]
[1]Beijing University of Posts and Telecommunications
[2]Beijing Academy of Artificial Intelligence
[3]Chinese Academy of Sciences
[4]University of Science and Technology of China
{cfli, shaoyx}@bupt.edu.cn   qinminghao24@ia.ac.cn
stxiao@baai.ac.cn   chenjianlv@mail.ustc.edu.cn
liandefu@ustc.edu.cn   {luokun695, zhengliu1026}@gmail.com

## ABSTRACT

Large language models (LLMs) with decoder-only architectures have demonstrated exceptional text-generation capabilities across a variety of tasks. Some researchers have also adapted these models for text representation tasks. However, in text representation tasks, these models often face performance degradation on unseen tasks. In-context learning (ICL), which leverages examples provided in the input context, enables LLMs to handle unseen tasks effectively. Inspired by this, we aim to fully utilize the inherent properties of LLMs to enhance text representation performance across different tasks through the ICL approach.

In this paper, we introduce a simple yet effective training strategy, which significantly improves text representation capabilities. Unlike previous models that prepend task instructions to the text, our method randomly samples a varying number of examples during training, endowing the embedding model with in-context learning abilities while maintaining its zero-shot capabilities. This approach does not require additional data construction or modifications to the model architecture. On the contrary, we find that some popular modifications to the model, such as bidirectional attention, can degrade performance, undermining the inherent characteristics of LLMs. We have publicly released our method at this repo.

## 1 INTRODUCTION

Text embeddings are vector representations that capture the semantic and contextual meaning of natural language text. They play a pivotal role in natural language processing (NLP) tasks, facilitating a wide range of applications such as information retrieval, text classification, item recommendation, and question answering (Karpukhin et al., 2020; Xiong et al., 2020; Lu et al., 2020; Zhou et al., 2024b). Pre-trained bidirectional encoder and encoder-decoder architectures have been widely adopted as backbone models for embedding model, owing to their effectiveness in producing high-quality vector embeddings for text thanks to their extensive pre-training (Xiao et al., 2022; Gao et al., 2021).

The impressive performance showcased by Large Language Models (LLMs) has sparked a growing interest in exploring how these decoder-only models can be utilized as embedding models (Ma et al., 2023; Li et al., 2024; Wang et al., 2023b). These LLM-based embedding models have exhibited remarkable enhancements in domain-specific accuracy and generalization capabilities, particularly when trained through supervised learning approaches (Wang et al., 2023b). A popular adaptation in training LLMs for embedding purposes is instruction-tuning (Wei et al., 2021; Ouyang et al., 2022), which involves providing varied instructions specific to different tasks. This targeted fine-tuning has proven superior to traditional methods (Wang et al., 2023b; Lee et al., 2024a; Asai et al., 2022; Wang et al., 2023a). However, the information contained within the instructions is still limited.

---

*Co-first authors
†Corresponding authors, with Zheng Liu as the project lead

| Instruction | Given a scene, retrieve the fairy tale that unfolds with this scene. |
|---|---|
| Examples | **Scene:** A young girl discovers an old, dusty book in an attic.
**Fairy Tale:** Once upon a time, a curious young girl named Eliza found an old, dusty book in her grandmother's attic. As she opened it, she was transported into a magical realm where she had to help a brave knight save a cursed kingdom. Together, they broke the curse and restored peace.

**Scene:** A frog is sitting on a lilypad under a moonlit sky.
**Fairy Tale:** Under a moonlit sky, a cursed prince in the form of a frog sat on a lilypad. A kind maiden named Lila came by and, moved by his sorrow, kissed him. The curse was broken, and the frog transformed into a prince. They married and ruled a kingdom happily ever after.

**Scene:** A cat is chasing a mouse through a castle.
**Fairy Tale:** In an ancient castle, a mouse named Max and a cat named Sir Whiskers stumbled upon a secret chamber with a magical crystal. Instead of continuing their chase, they called a truce to protect the crystal. Together, they used its magic to bring prosperity and harmony to the castle. |
| Query | A group of rabbits are running. |
| Candidates | **1** On a meadow, a group of rabbits are running, with an eagle chasing them from behind. To survive, the rabbits must run as fast as they can.

**2** Once upon a time, in a blooming meadow, a group of rabbits were happily racing each other. Their playful chase led them to a hidden, glowing burrow. Inside, they discovered an enchanted world where animals spoke and wishes came true, a secret haven of endless adventures. |
| **Similarity Score** | **e5-mistral** **1** > **2** ✗  **gte-qwen2** **1** > **2** ✗  **bge-en-icl (zero-shot)** **1** > **2** ✗  **bge-en-icl (few-shot)** **2** > **1** ✓ |

Figure 1: For a new task, conventionally tuned instruction-tuning embedding models use only task instructions and queries. They assign a higher similarity score to the incorrect candidate 1 than to the correct candidate 2, resulting in incorrect retrieval results. However, with our ICL strategy-trained embedding model, although zero-shot retrieval results may remain incorrect, providing a few examples enables the model to retrieval successfully.

When faced with an entirely new task, the model may struggle to fully understand the task based solely on the instructions. For example, as shown in Figure 1, when given a new retrieval task, both E5-mistral (Wang et al., 2023b) and GTE-qwen2 (Li et al., 2023) assign a higher similarity score to the incorrect candidate 1 than to the correct candidate 2, resulting in incorrect retrieval results.

In-context learning (ICL) is a core capability of LLMs, enabling them to incorporate task-specific examples directly into input prompts to generate desired outputs (Radford et al., 2019; Brown, 2020; Gao et al., 2020). The scope of ICL extends beyond tasks seen during training; it enables LLMs to generalize to new and complex tasks by learning patterns from the provided examples. This allows LLMs to adapt dynamically to novel tasks without additional training, making them highly applicable to real-world scenarios (Wei et al., 2022; Yao et al., 2022; Dong et al., 2022; Zhou et al., 2024c).

Recognizing the robust ICL abilities of LLMs, in this work, we introduce ICL Embedder, a model capable of handling various tasks within a single framework by given the input text, task instruction and a few task-related examples. Unlike previous models, we not only provide task instructions to guide the generation of query embeddings but also incorporate task-related examples to further enhance the query embeddings. To train the ICL Embedder, we randomly select examples for each training step to ensure robust few-shot capabilities, and we use a diverse numbers of examples to train for maintaining the model's zero-shot performance. As illustrated in Figure 1, while our model *bge-en-icl* exhibits unsatisfactory performance in the zero-shot scenario, its retrieval accuracy significantly improve when provided with few-shot examples. To the best of our knowledge, this is the first embedding model to leverage the ICL strategy for generating embeddings. Our model *bge-en-icl* achieves state-of-the-art (SOTA) results on both the MTEB (Up to August 29, 2024) (Muennighoff et al., 2022) and AIR-Bench (Chen et al., 2024) benchmarks.

Moreover, LLMs are predominantly utilized for text generation tasks, and adapting them for text representation tasks requires specific fine-tuning strategies. Recent studies have introduced various approaches, including the generation of high-quality training data through LLMs (Wang et al., 2023b), modifications to attention mechanisms, and changes in pooling methods (Ma et al., 2023; Li et al., 2024). In this paper, we also investigate how to effectively utilize LLMs as embedding models by modifying various architectures, e.g., bidirectional attention, meaning pooling. Our experimental

findings indicate that in the ICL scenario, making complex modifications to the models does not lead to significant improvements. In contrast, the best results are obtained using the original, unmodified architecture.

In summary, the key contributions of our work are as follows:

- We propose to integrate ICL capabilities into the embedding model and introduce a simple but effective training strategy, which empowers the ICL Embedder to achieve exceptional performance without requiring additional training data or modifications to the model architecture. Remarkably, our model *bge-en-icl* achieves SOTA performance on both the MTEB and AIR-Bench benchmarks. To the best of our knowledge, this is the first work to successfully incorporate ICL capabilities into an embedding model.
- We rethink and explore how to effectively utilize LLMs as embedding models by evaluating various attention mechanisms and pooling methods. Our findings highlight that simplicity is best; simply combining ICL capabilities with embedding models can achieve excellent performance.
- In contrast to other leading models on the MTEB benchmark, we provide open access to our model checkpoint, dataset, and training scripts.

## 2 RELATED WORK

Text embedding is a critical research direction in the field of information retrieval, with wide-ranging applications including web search, question answering, and dialogue systems. (Fujiwara et al., 2023; Jiajia WANG, 2023; Yuan GAO, 2023) The fundamental principle involves encoding both queries and documents into embedding vectors within the same latent space. By calculating similarity scores between these vectors, effective retrieval is achieved. In recent years, numerous studies have leveraged pre-trained language models such as BERT (Devlin, 2018), T5 (Raffel et al., 2020), and RoBERTa (Liu, 2019) as the backbone for embedding models. They have consistently demonstrated superior performance compared to sparse retrieval methods.

The capability of the backbone is a crucial determinant in the effectiveness of retrieval systems. (Luo et al., 2024) have demonstrated that performance improves with increased scale and extensive pre-training. Currently, numerous studies have explored the effectiveness of utilizing LLMs as backbone encoders for text embedding tasks.

Repllama (Ma et al., 2023) fine-tuned Llama-2 to serve as both a dense retriever and a reranker, demonstrating the effectiveness of applying large language models (LLMs) in text embedding tasks. To further align LLMs with text embedding tasks, Llama2Vec (Li et al., 2024) introduced two pre-training tasks specifically designed to enhance the model's performance, which led to significant improvements on the BEIR benchmark. E5-mistral and Gecko (Wang et al., 2023b; Lee et al., 2024b) advanced the training of LLM-based embedding models through the use of synthetic data (Zhou et al., 2024a), markedly boosting their performance across a diverse range of retrieval and non-retrieval tasks. NV-Embed (Lee et al., 2024a) innovatively proposed a latent attention layer to replace conventional pooling methods and implemented a two-stage training strategy to address the challenge of false negatives in non-retrieval tasks. This model has shown strong performance in both retrieval and non-retrieval domains. Additionally, GRIT (Muennighoff et al., 2024) success-fully integrated text embedding and generation within a single LLM, achieving performance levels on par with specialized models focused solely on either embedding or generation. In the exploration of LLMs as embedding models from an unsupervised perspective, LLM2Vec (BehnamGhader et al., 2024) presented a novel unsupervised method to transform decoder-only LLMs into embedding models. This approach demonstrated significant potential for modifying LLM backbone encoders to perform retrieval without any supervision. Similarly, PromptReps (Zhuang et al., 2024) leveraged chat-based LLMs aligned with human preferences to generate high-quality dense representations in an unsupervised manner.

The LLM-based embedding models mentioned above exhibit commendable performance across both retrieval and non-retrieval tasks. However, much of the existing work has disproportionately focused on altering model architectures, thereby neglecting the intrinsic capabilities of LLMs. Even models like GritLM, which integrate generation and embedding functionalities, fail to fully exploit the potential ICL capabilities of LLMs within the embedding process. By leveraging the innate

Figure 2: The query representation of the ICL Embedder.

ICL capabilities of LLMs, embedding models can be more versatile and adapt to diverse scenarios without necessitating additional fine-tuning. Our model effectively utilizes the inherent strengths of LLMs and achieves SOTA results on the MTEB and AIR-Bench benchmarks.

## 3 METHOLOGY

To ensure the embedding models can be used for various tasks with ICL capabilities, without any additional training, we propose a simple yet effective strategy. We will present the ICL representation for embedding models and our training strategy in the following section.

### 3.1 THE ICL REPRESENTATION FOR EMEBDDING MODELS

Traditional embedding models often directly input the query to generate target embeddings. However, this approach struggles to handle tasks with different intents, limiting the model's adaptability and generalization capabilities. To overcome this limitation, researchers have introduced appending task instructions (Su et al., 2022) to queries, enabling a single embedding model to generalize across various domains by altering the instructions.

Despite these advancements, the information provided by the instruction remains constrained. Inspired by the remarkable capability of LLMs to adapt and perform well on unseen tasks through ICL, we seek to integrate this powerful feature into our embedding model. Consequently, we propose an innovative query representation format that leverages ICL, as depicted in Figure 2.

Consider a new query $q^+$, its corresponding positive passage $p^+$, and a few-shot set of $n$ query-passage pairs $\{(q_1, p_1), \ldots, (q_n, p_n)\}$ in an embedding task. The traditional instruction-based query template (Wang et al., 2023b) has the following format:

$$\langle \text{Instruct} \rangle \ \{ \text{task\_definition} \} \ \langle \text{query} \rangle \ \{ q^+ \} \tag{1}$$

Here, "*task\_definition*" represents the description of the specific embedding task. However, the information provided in the instruction alone is also limited. To overcome this limitation, we propose an expanded query format that incorporates few-shot examples. First, we suggest organizing each query-passage pair $(q_k, p_k)$ as follows:

$$\langle \text{Instruct} \rangle \ \{ \text{task\_definition} \} \ \langle \text{query} \rangle \ \{ q_k \} \ \langle \text{response} \rangle \ \{ p_k \} \tag{2}$$

Once the few-shot examples are obtained, they can be concatenated with the query to form the following format:

$$\{\text{example 1}\} ... \{\text{example n}\} \ \langle\text{Instruct}\rangle \ \{\text{task\_definition}\} \ \langle\text{query}\rangle \ \{q^+\} \ \langle\text{response}\rangle \quad (3)$$

We then append an [EOS] token to the end of the modified input queries and passages, and feed them into the language model to obtain embeddings $(h_{q^+}, h_{p^+})$, the final hidden state of the [EOS] token is used as the embedding, and we apply the same [EOS] token for encoding both queries and passages.

## 3.2 THE ICL EMBEDDER TRAINING STRATEGY

While previous works (Wang et al., 2023b; Lee et al., 2024a) have proposed the training method of instruction-tuning, which incorporates a large number of task-specific instructions during the training process, enabling the model to adapt to various downstream retrieval tasks based on different instructions, it is not applicable to the ICL strategy. As demonstrated by GRIT (Muennighoff et al., 2024), directly supplying few-shot examples when generating embeddings can actually degrade model performance.

A straightforward approach to train ICL Embedder is providing task-specific few-shot examples and instructions along with each query during training, which helps the model effectively leverage the examples to enhance the representation of the query's embedding. However, such a training process raises several issues. On the one hand, if few-shot samples are always used during the training process, there is a risk that the model's zero-shot capabilities could be hindered. On the other hand, if the examples used for training remain static, the model may not be able to handle new examples, resulting in a decline in performance in the few-shot scenario.

To enable the embedding model with ICL capabilities and ensure high performance in both zero-shot and few-shot scenarios, we propose a simple yet effective training strategy. Within each training batch, we utilize the same dataset. During the training process, we select different examples from the same batch at each step to ensure variability in the data exposed to the model, thereby enhancing its few-shot robustness. Simultaneously, the number of examples is randomly chosen between 0 and the maximum value, which supports the development of the model's zero-shot capabilities.

During training, we employ the standard InfoNCE (Izacard et al., 2021) loss function $\mathcal{L}$:

$$\mathcal{L} = -\log \frac{\exp(\text{sim}(q^+, p^+))}{\exp(\text{sim}(q^+, p^+)) + \sum_j \exp(\text{sim}(q^+, p_j^-))} \quad (4)$$

In this equation, $p_j^-$ denotes the set of negative passages. For retrieval tasks, this set encompasses both in-batch negatives and hard negatives, whereas for non-retrieval tasks, it is limited solely to hard negatives. The function $\text{sim}(q, p)$ is the scoring function between the query and passage. The scoring function is a temperature-scaled cosine similarity, defined as:

$$\text{sim}(q, p) = \frac{1}{\tau} \cos(h_q, h_p) \quad (5)$$

Here, $\tau$ is a temperature hyperparameter, which is fixed at 0.02 during training. The $\cos(h_q, h_p)$ term represents the cosine similarity between the query representation $h_q$ and passage representations $h_p$.

## 4 EXPERIMENTS

In this section, we examine the effectiveness of the ICL Embedder training strategy and rethink the training methodologies for LLM-based embedding models. We focus on the following questions:

- **RQ 1:** How does our ICL Embedder perform in zero-shot and few-shot scenarios?
- **RQ 2:** How does the performance of our ICL Embedder compare to other LLM-based embedding methods?
- **RQ 3:** How does our ICL training strategy affect the performance of embedding models compared to normal ICL training strategy.
- **RQ 4:** Will changes in model architecture, such as bidirectional attention and mean pooling, improve the performance of ICL Embedder?

## 4.1 SETUP

**LLM.** Following E5-Mistral (Wang et al., 2023b), SFR, and NV-Embedder (Lee et al., 2024a), we have adopted Mistral-7B (Jiang et al., 2023) as the backbone for our framework.

**Training Data.** To ensure a fair comparison, we use the **E5-Mistral dataset**, which is employed to fine-tune both the E5-Mistral (Wang et al., 2023b) and LLM2Vec (BehnamGhader et al., 2024). This dataset includes some in-domain retrieval datasets from MTEB, including HotpotQA (Yang et al., 2018), FEVER (Thorne et al., 2018), MSMARCO passage ranking (Nguyen et al., 2016), NQ (Karpukhin et al., 2020) and Quora Duplicate Questions (DataCanary et al., 2017), as well as other publicly available retrieval datasets, including ELI5 (Fan et al., 2019), MIRACL (Zhang et al., 2023), MSMARCO document ranking (Nguyen et al., 2016), NLI (Gao et al., 2021), SQuAD (Karpukhin et al., 2020), TriviaQA (Karpukhin et al., 2020), MrTyDi (Zhang et al., 2021), DuReader (Qiu et al., 2022), and T2Ranking.

However, methods that typically perform exceptionally well, such as NV-Embedder (Lee et al., 2024a) and SFR, often require more MTEB in-domain training data. Additionally, some of these methods, such as GTE-Qwen2 (Li et al., 2023), do not disclose their sources of training data. We speculate that they might have also utilized additional MTEB in-domain data. Therefore, we have collected a new dataset, the **Augmented E5-Mistral dataset**. This dataset builds on the English retrieval dataset from the E5-Mistral dataset, incorporating extra in-domain training data from MTEB. It includes data for various tasks including Retrieval, Reranking, Clustering, Classification, and STS. Specifically, the Augmented E5-Mistral dataset contains the following categories of datasets:

- **Retrieval**: ELI5, HotpotQA, FEVER, MSMARCO passage and document ranking, NQ, NLI, SQuAD, TriviaQA, Quora Duplicate Questions, Arguana (Wachsmuth et al., 2018), and FiQA (Maia et al., 2018).

- **Reranking**: SciDocsRR (Cohan et al., 2020) and StackOverFlowDupQuestions (Liu et al., 2018).

- **Classification**: AmazonReviews-Classification (McAuley & Leskovec, 2013), Amazon-Counterfactual-Classification (O'Neill et al., 2021), Banking77-Classification (Casanueva et al., 2020), Emotion-Classification (Saravia et al., 2018), TweetSentimentExtraction-Classification (Maggie, 2020), MTOPIntent-Classification (Li et al., 2020), IMDB-Classification (Maas et al., 2011), ToxicConversations-Classification (Adams et al., 2019).

- **Clustering**: {Arxiv/Biorxiv/Medrxiv/Reddit/StackExchange}-Clustering-{S2S/P2P}, TwentyNewsgroups-Clustering (Lang, 1995).

- **STS**: STS12 (Agirre et al., 2012), STS22 (Chen et al., 2022), STS-Benchmark (Cer et al., 2017).

**Training Detail.** We fine-tune the Mistral-7B model using the contrastive loss and train it for a single epoch. For efficient fine-tuning, we employ Low-Rank Adaptation (LoRA) (Hu et al., 2021), setting the LoRA rank to 64 and the LoRA alpha to 32, with a learning rate of 1e-4. For retrieval tasks, we use in-batch negatives. Each dataset incorporates 7 hard negatives. The batch size is set to 512 for retrieval tasks and 256 for other types of tasks. We maintain consistency by using the same dataset throughout one training step. To distill the score from reranker in retrieval tasks, we use the bge-reranker model (Liu et al., 2025) as the teacher. For in-context learning training, we implement a in-batch random examples selection training strategy. For each query, considering excessively long inputs will severely restrict the batch size, we select between 0 to 5 examples from the in-batch training data. In training, the maximum length for the query, passage, and example is set to 512. The example comprises the example query and example passage, each with a maximum length of 256. The maximum length for the concatenated query and examples is 2048.

**Evaluation.** We evaluate the performance of our model on MTEB (Muennighoff et al., 2022) and AIR-Bench (Chen et al., 2024). MTEB is a comprehensive benchmark designed to evaluate the performance of text embedding models. AIR-Bench is dedicated to the evaluation of retrieval performance, its testing data is automatically generated by large language models without human intervention. We evaluate the performance of our model under both zero-shot and few-shot scenarios. In the few-shot scenario, fixed in-context examples are applied to each query within the same dataset. We use the following strategy to select examples for evaluation:

| Task | Retr. | Rerank. | Clust. | PairClass. | Class. | STS | Summ. | Avg. |
|---|---|---|---|---|---|---|---|---|
| # of datasets → | 15 | 4 | 11 | 3 | 12 | 10 | 1 | 56 |
| w/ E5-Mistral dataset | | | | | | | | |
| E5-mistral-7b-instruct | 52.78 | 60.38 | 47.78 | 88.47 | 76.80 | 83.77 | 31.90 | 64.56 |
| GritLM-7B | 53.10 | 61.30 | 48.90 | 86.90 | 77.00 | 82.80 | 29.40 | 64.70 |
| LLM2Vec-Mistral-supervised | 55.99 | 58.42 | 45.54 | 87.99 | 76.63 | 84.09 | 29.96 | 64.80 |
| **bge-en-icl (E5-Mistral dataset) (zero-shot)** | 59.59 | 56.85 | 42.61 | 87.87 | 75.47 | 83.30 | 29.52 | 64.67 |
| **bge-en-icl (E5-Mistral dataset) (few-shot)** | 60.08 | 56.67 | 46.55 | 88.51 | 77.31 | 83.69 | 30.68 | **66.08** |
| E5-mistral-7b-instruct | 56.90 | 60.21 | 50.26 | 88.34 | 78.47 | 84.66 | 31.40 | 66.63 |
| GritLM-7B | 57.41 | 60.49 | 50.61 | 87.16 | 79.46 | 83.35 | 30.37 | 66.76 |
| SFR-Embedding | 59.00 | 60.64 | 51.67 | 88.54 | 78.33 | 85.05 | 31.16 | 67.56 |
| Linq-Embed-Mistral | 60.19 | 60.29 | 51.42 | 88.35 | 80.20 | 84.97 | 30.98 | 68.17 |
| voyage-large-2-instruct | 58.28 | 60.09 | 53.35 | 89.24 | 81.49 | 84.31 | 30.84 | 68.23 |
| NV-Embed-v1 | 59.36 | 60.59 | 52.80 | 86.91 | 87.35 | 82.84 | 31.20 | 69.32 |
| bge-multilingual-gemma2 | 59.24 | 59.72 | 54.65 | 85.84 | 88.08 | 83.88 | 31.20 | 69.88 |
| stella_en_400M_v5 | 58.97 | 60.16 | 56.70 | 87.74 | 86.67 | 84.22 | 31.66 | 70.11 |
| gte-Qwen2-7B-instruct | 60.25 | 61.42 | 56.92 | 85.79 | 86.58 | 83.04 | 31.35 | 70.24 |
| SFR-Embedding-2_R | 60.18 | 60.14 | 56.17 | 88.07 | 89.05 | 81.26 | 30.71 | 70.31 |
| stella_en_1.5B_v5 | 61.01 | 61.21 | 57.69 | 88.07 | 87.63 | 84.51 | 31.49 | 71.19 |
| NV-Embed-v2 (August 30, 2024) | 62.65 | 60.65 | 58.46 | 88.67 | 90.37 | 84.31 | 30.70 | **72.31** |
| **bge-en-icl (Augmented E5-Mistral dataset) (zero-shot)** | 61.67 | 59.66 | 57.51 | 86.93 | 88.62 | 83.74 | 30.75 | 71.24 |
| **bge-en-icl (Augmented E5-Mistral dataset) (few-shot)** | 62.16 | 59.82 | 57.89 | 88.14 | 88.95 | 84.24 | 30.77 | **71.67** |

Table 1: The performance on the MTEB benchmark.

| Domain | wiki | web | news | healthcare | law | finance | arxiv | msmarco | Avg. |
|---|---|---|---|---|---|---|---|---|---|
| # of datasets → | 1 | 1 | 1 | 1 | 1 | 1 | 1 | 1 | 8 |
| E5-mistral-7b-instruct | 61.67 | 44.41 | 48.18 | 56.32 | 19.32 | 54.79 | 44.78 | 59.03 | 48.56 |
| SFR-Embedding | 63.46 | 51.27 | 52.21 | 58.76 | 23.27 | 56.94 | 47.75 | 58.99 | 51.58 |
| NV-Embed-v1 | 62.84 | 50.42 | 51.46 | 58.53 | 20.65 | 49.89 | 46.10 | 60.27 | 50.02 |
| Linq-Embed-Mistral | 61.04 | 48.41 | 49.44 | 60.18 | 20.34 | 50.04 | 47.56 | 60.50 | 49.69 |
| gte-Qwen2-7B-instruct | 63.46 | 51.20 | 54.07 | 54.20 | 22.31 | 58.20 | 40.27 | 58.39 | 50.26 |
| stella_en_1.5B_v5 | 61.99 | 50.88 | 53.87 | 58.81 | 23.22 | 57.26 | 44.81 | 61.38 | 51.53 |
| NV-Embed-v2 (August 30, 2024) | 65.19 | 52.58 | 53.13 | 59.56 | 25.00 | 53.04 | 48.94 | 60.80 | 52.28 |
| **bge-en-icl (Augmented E5-Mistral dataset) (zero-shot)** | 64.61 | 54.40 | 55.11 | 57.25 | 25.10 | 54.81 | 48.46 | 63.71 | 52.93 |
| **bge-en-icl (Augmented E5-Mistral dataset) (few-shot)** | 64.94 | 55.11 | 56.02 | 58.85 | 28.29 | 57.16 | 50.04 | 64.50 | **54.36** |
| **bge-en-icl (E5-Mistral dataset) (zero-shot)** | 64.82 | 54.96 | 55.82 | 57.06 | 28.87 | 54.46 | 49.60 | 63.25 | 53.60 |
| **bge-en-icl (E5-Mistral dataset) (few-shot)** | 66.98 | 56.38 | 57.17 | 59.54 | 32.03 | 58.81 | 51.36 | 65.05 | **55.92** |

Table 2: QA (en, nDCG@10) performance on AIR-Bench.

- **For tasks with training sets**: We reserve a small subset of the training set for testing purposes. From the remaining training data, we randomly sample examples three times. We then select the set of examples that achieves the highest accuracy on the small subset as the final in-context examples for evaluation.

- **For tasks without training sets**: We provide ChatGPT with a task description to generate 10 examples. Then, we use ChatGPT to filter these examples and select those that best represent the task.

## 4.2 MAIN RESULTS (FOR RQ 1 AND RQ 2)

**MTEB.** Table 1 presents the performance of our model, *bge-en-icl*, evaluated on the MTEB benchmark. It is important to note that the use of the Augmented E5-Mistral dataset may introduce unfair comparisons, as different models often rely on varying datasets, and many of these models do not disclose the specific datasets they use. For a fairer comparison and to better understand the impact of in-context learning, we conducts an evaluation using the E5-Mistral dataset. Under these constraints, our model's performance in the zero-shot scenario is on par with that of other models such as GritLM (Muennighoff et al., 2024) and LLM2Vec (BehnamGhader et al., 2024). However, in the few-shot scenario, our model show significant enhancements, particularly in the classification and clustering tasks that are not part of the training data. These improvements underscore the potential benefits of in-context learning, demonstrating its generalizability and effectiveness when applied to tasks outside the original training domain.

When leveraging the Augmented E5-Mistral dataset, our model demonstrates strong capabilities in both zero-shot and few-shot scenarios, achieving SOTA results in the few-shot scenario (Up to August 29, 2024). However, the performance in the few-shot scenario exhibits only a marginal improvement over the zero-shot scenario. Because the model is previously exposed to these specific datasets during its training phase, enables it to perform well on their corresponding test sets. As a result, employing ICL does not yield significant benefits.

| Domain
# of datasets → | arxiv
4 | book
2 | healthcare
5 | law
4 | Avg.
15 |
|---|---|---|---|---|---|
| text-embedding-3-large | 74.53 | 73.16 | 65.83 | 64.47 | 68.77 |
| E5-mistral-7b-instruct | 72.14 | 72.44 | 68.44 | 62.92 | 68.49 |
| SFR-Embedding | 72.79 | 72.41 | 67.94 | 64.83 | 69.00 |
| NV-Embed-v1 | 77.65 | 75.49 | 72.38 | 69.55 | 73.45 |
| Linq-Embed-Mistral | 75.46 | 73.81 | 71.58 | 68.58 | 72.11 |
| gte-Qwen2-7B-instruct | 63.93 | 68.51 | 65.59 | 65.26 | 65.45 |
| stella_en_1.5B_v5 | 73.17 | 74.38 | 70.02 | 69.32 | 71.25 |
| bge-multilingual-gemma2 | 71.77 | 76.46 | 73.96 | 70.86 | 72.88 |
| NV-Embed-v2 (August 30, 2024) | 79.27 | 77.46 | 73.01 | 71.18 | 74.78 |
| **bge-en-icl (Augmented E5-Mistral dataset) (zero-shot)** | 78.30 | 78.21 | 73.65 | 67.09 | 73.75 |
| **bge-en-icl (Augmented E5-Mistral dataset) (few-shot)** | 79.63 | 79.36 | 74.80 | 67.79 | **74.83** |
| **bge-en-icl (E5-Mistral dataset) (zero-shot)** | 79.73 | 78.66 | 72.88 | 70.59 | 74.86 |
| **bge-en-icl (E5-Mistral dataset) (few-shot)** | 79.82 | 80.37 | 74.60 | 71.66 | **75.98** |

Table 3: Long-Doc (en, Recall@10) performance on AIR-Bench.

**AIR-Bench.** Our model's performance is further evaluated on the AIR-Bench dataset, encompassing QA and Long-Doc tasks. As shown in Tables 2 and 3, our model demonstrates significant performance across both QA and Long-Doc tasks when trained on either the Augmented E5-Mistral dataset or the E5-Mistral dataset. It is noteworthy that there is no overlap between the model's training dataset and the AIR-Bench evaluation data, and our model's few-shot performance significantly surpasses its zero-shot performance in all cases, underscoring its robustness in handling unseen tasks.

Interestingly, the model achieves better results when trained solely on the E5-Mistral dataset compared to training on the Augmented E5-Mistral dataset. This improvement could be attributed to that the Augmented E5-Mistral dataset containing an excessive amount of MTEB-related data, such as clustering and classification tasks. Such data might introduce the risk of overfitting, thereby potentially hampering the model's generalization performance on the AIR-Bench dataset.

## 4.3 IN-CONTEXT LEARNING (FOR RQ 3)

| Task
# of datasets → | Retr.
15 | Rerank.
4 | Clust.
11 | PairClass.
3 | Class.
12 | STS
10 | Summ.
1 | Avg.
56 |
|---|---|---|---|---|---|---|---|---|
| w/ E5-Mistral dataset | | | | | | | | |
| w/o in-context learning (zero-shot) | 59.11 | 57.02 | 42.60 | 87.99 | 76.27 | 83.93 | 30.50 | 64.83 |
| w/ fix examples (zero-shot) | 48.98 | 56.48 | 41.84 | 85.94 | 74.38 | 84.31 | 29.68 | 61.50 |
| w/ fix examples (few-shot) | 59.00 | 56.90 | 45.75 | 88.54 | 75.56 | 84.67 | 30.66 | 65.46 |
| w/ random examples (zero-shot) | 59.59 | 56.85 | 42.61 | 87.87 | 75.47 | 83.30 | 29.52 | 64.67 |
| w/ random examples (few-shot) | 60.08 | 56.67 | 46.55 | 88.51 | 77.31 | 83.69 | 30.68 | **66.08** |

Table 4: Evaluation of various ICL strategies on the MTEB Benchmark.

To evaluate the impact of the ICL strategy, we conduct a series of ablation studies using the MTEB benchmark. In these studies, we compare the performance of models fine-tuned with the ICL strategy against those fine-tuned without it. These experiments all use the same setting. Specifically, for ICL training, we employ two distinct training approaches: fixed examples and random examples. For fixed examples, each task is trained using three predetermined examples. For random examples, constitution and quantity of examples are random.

Table 4 presents various results from our experiment. When trained without the ICL strategy, the model's zero-shot performance is 64.83. However, its performance significantly degrades and can even become unusable when provided with few-shot examples.

When fixed examples are used during ICL training, there is a significant decline in zero-shot evaluation performance compared to using random examples. This decline can be attributed to the model's consistent exposure to the same training examples, which may have impaired its zero-shot abilitiy. On the other hand, in the few-shot scenario, the model demonstrates improved performance when trained with fixed examples, exceeding its own zero-shot results by 3.96 points and outperforming models trained without ICL by 0.63 points. This confirms the effectiveness of the ICL strategy in enhancing model performance.

When utilizing random examples during training, the model's zero-shot capability is preserved. Furthermore, exposing the model to random examples enhances its performance in the few-shot scenario due to the abundance of examples it encounters during the training process.

## 4.4 ATTENTION (FOR RQ 4)

| Task
# of datasets → | Retr.
15 | Rerank.
4 | Clust.
11 | PairClass.
3 | Class.
12 | STS
10 | Summ.
1 | Avg.
56 |
|---|---|---|---|---|---|---|---|---|
| causal attention & last token pooling | | | | | | | | |
| w/o in-context learning | 59.11 | 57.02 | 42.60 | 87.99 | 76.27 | 83.93 | 30.50 | 64.83 |
| w/ in-context learning (zero-shot) | 59.59 | 56.85 | 42.61 | 87.87 | 75.52 | 83.30 | 29.52 | 64.67 |
| w/ in-context learning (few-shot) | 60.08 | 56.67 | 46.55 | 88.51 | 77.31 | 83.69 | 30.68 | **66.08** |
| causal attention & mean pooling | | | | | | | | |
| w/o in-context learning | 58.50 | 53.74 | 36.82 | 82.14 | 72.37 | 77.62 | 29.10 | 61.03 |
| bidirectional attention & last token pooling | | | | | | | | |
| w/o in-context learning | 59.59 | 56.96 | 44.34 | 87.61 | 74.77 | 83.81 | 30.12 | 64.96 |
| w/ in-context learning (zero-shot) | 59.77 | 58.09 | 44.04 | 87.87 | 75.35 | 83.97 | 29.75 | 65.19 |
| w/ in-context learning (few-shot) | 60.23 | 57.81 | 44.45 | 88.64 | 77.00 | 83.77 | 29.99 | **65.74** |
| bidirectional attention & mean pooling | | | | | | | | |
| w/o in-context learning | 59.13 | 57.03 | 43.44 | 87.25 | 75.03 | 84.08 | 29.17 | 64.73 |
| w/ in-context learning (zero-shot) | 59.53 | 57.48 | 43.88 | 88.12 | 74.86 | 83.64 | 29.58 | 64.90 |
| w/ in-context learning (few-shot) | 59.42 | 57.29 | 44.93 | 88.36 | 75.26 | 83.75 | 29.60 | **65.18** |

Table 5: Results of different attention and pooling mechanisms on the MTEB Benchmark.

Recent studies have explored modifying the attention mechanism in LLMs to adopt bidirectional attention and employ mean pooling for embedding generation. Notably, models such as GritLM (Muennighoff et al., 2024), NV-Embed (Lee et al., 2024a), and LLM2Vec (BehnamGhader et al., 2024) have successfully utilized these techniques, achieving considerable experimental success. Motivated by these advancements, we explore the potential benefits of implementing bidirectional attention in the ICL scenario. Specifically, we investigate the impacts of various attention and pooling mechanisms, including causal and bidirectional attention, coupled with last token pooling and mean pooling.

In a causal attention framework, each token is limited to accessing only the information from preceding tokens, without considering subsequent tokens, and employing mean pooling tends to yield bad results due to this restriction. Therefore, in this specific configuration, we present only the results from experiments without ICL.

Table 5 presents the experimental setup and results in both non-ICL and ICL scenarios. It shows that in non-ICL scenarios, most methods yield consistent performance, except for the combination of causal attention with mean pooling. In contrast, in ICL scenarios, the integration of causal attention and last token pooling emerges as the superior approach. This configuration seems aligned to the model's pre-training, suggesting that retaining the original architecture and simplicity is advantageous. Moreover, shifting from causal attention to bidirectional attention does not lead to significant improvements, and mean pooling is not necessary for implementing bidirectional attention.

Additionally, configurations utilizing bidirectional attention paired with last token pooling are also effective in both non-ICL and zero-shot scenarios, indicating that it is a viable option in some specific scenarios.

## 5 CONCLUSION

This paper proposes a novel approach that enables embedding models to leverage ICL capabilities without requiring additional data or modifications to the model architecture. To the best of our knowledge, this is the first work to successfully apply ICL capabilities to embedding models through a simple yet effective training strategy. Our approach empowers embedding models to become in-context learners, and experimental results demonstrate that our model achieves SOTA performance on the MTEB and AIR-Bench datasets.

Furthermore, we rethink and explore potential changes to the model structure, such as bidirectional attention. Our findings indicate that these structural modifications do not enhance the few-shot performance of the embedding models but instead lead to a decline in performance. We hope that the ICL Embedder could provide valuable insights for both researchers and practitioners working with embedding models and in-context learning.

# 6 ACKNOWLEDGEMENTS

This work is supported by the National Science and Technology Major Project (2023ZD0121504), the National Science and Technology Major Project (2022ZD0116315), National Natural Science Foundation of China (Nos. 62272054, 62192784, U24A20253), Beijing Nova Program (No. 20230484319), and Xiaomi Young Talents Program.

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

# A  INSTRUCTION

| Task Name | Instruction Template |
|---|---|
| ArguAna | Given a claim, find documents that refute the claim. |
| ClimateFEVER | Given a claim about climate change, retrieve documents that support or refute the claim. |
| CQADupStack | Given a question, retrieve detailed question descriptions from Stackexchange that are duplicates to the given question. |
| DBPedia | Given a query, retrieve relevant entity descriptions from DBPedia. |
| FEVER | Given a claim, retrieve documents that support or refute the claim. |
| FiQA2018 | Given a financial question, retrieve user replies that best answer the question. |
| HotpotQA | Given a multi-hop question, retrieve documents that can help answer the question. |
| MSMARCO | Given a web search query, retrieve relevant passages that answer the query. |
| NFCorpus | Given a question, retrieve relevant documents that best answer the question. |
| Natural Question | Given a question, retrieve Wikipedia passages that answer the question. |
| QuoraRetrieval | Given a question, retrieve questions that are semantically equivalent to the given question. |
| SCIDOCS | Given a scientific paper title, retrieve paper abstracts that are cited by the given paper. |
| SciFact | Given a scientific claim, retrieve documents that support or refute the claim. |
| Touche2020 | Given a question, retrieve detailed and persuasive arguments that answer the question. |
| TREC-COVID | Given a query, retrieve documents that answer the query. |
| STS* | Retrieve semantically similar text. |
| SummEval | Given a news summary, retrieve other semantically similar summaries. |
| AmazonCounterfactualClassification | Classify a given Amazon customer review text as either counterfactual or not-counterfactual. |
| AmazonPolarityClassification | Classify Amazon reviews into positive or negative sentiment. |
| AmazonReviewsClassification | Classify the given Amazon review into its appropriate rating category. |
| Banking77Classification | Given a online banking query, find the corresponding intents. |
| EmotionClassification | Classify the emotion expressed in the given Twitter message into one of the six emotions: anger, fear, joy, love, sadness, and surprise. |
| ImdbClassification | Classify the sentiment expressed in the given movie review text from the IMDB dataset. |
| MassiveIntentClassification | Given a user utterance as query, find the user intents. |
| MassiveScenarioClassification | Given a user utterance as query, find the user scenarios. |
| MTOPDomainClassification | Classify the intent domain of the given utterance in task-oriented conversation. |
| MTOPIntentClassification | Classify the intent of the given utterance in task-oriented conversation. |
| ToxicConversationsClassification | Classify the given comments as either toxic or not toxic. |
| TweetSentimentExtractionClassification | Classify the sentiment of a given tweet as either positive, negative, or neutral. |
| ArxivClusteringP2P | Identify the main and secondary category of Arxiv papers based on the titles and abstracts. |
| ArxivClusteringS2S | Identify the main and secondary category of Arxiv papers based on the titles. |
| BiorxivClusteringP2P | Identify the main category of Biorxiv papers based on the titles and abstracts. |
| BiorxivClusteringS2S | Identify the main category of Biorxiv papers based on the titles. |
| MedrxivClusteringP2P | Identify the main category of Medrxiv papers based on the titles and abstracts. |
| MedrxivClusteringS2S | Identify the main category of Medrxiv papers based on the titles. |
| RedditClustering | Identify the topic or theme of Reddit posts based on the titles. |
| RedditClusteringP2P | Identify the topic or theme of Reddit posts based on the titles and posts. |
| StackExchangeClustering | Identify the topic or theme of StackExchange posts based on the titles. |
| StackExchangeClusteringP2P | Identify the topic or theme of StackExchange posts based on the given paragraphs. |
| TwentyNewsgroupsClustering | Identify the topic or theme of the given news articles. |
| AskUbuntuDupQuestions | Retrieve duplicate questions from AskUbuntu forum. |
| MindSmallReranking | Retrieve relevant news articles based on user browsing history. |
| SciDocsRR | Given a title of a scientific paper, retrieve the titles of other relevant papers. |
| StackOverflowDupQuestions | Retrieve duplicate questions from StackOverflow forum. |
| SprintDuplicateQuestions | Retrieve duplicate questions from Sprint forum. |
| TwitterSemEval2015 | Retrieve tweets that are semantically similar to the given tweet. |
| TwitterURLCorpus | Retrieve tweets that are semantically similar to the given tweet. |
| AIR-Bench | Given a question, retrieve passages that answer the question. |

Table 6: The instruction we used on the MTEB and AIR-Bench benchmarks.

# B    DETAILED MTEB RESULTS

| Dataset | NV-Embed-v1 | bge-multilingual-gemma2 | gte-Qwen2-7B-instruct | SFR-Embedding-2_R | stella_en_1.5B_v5 | bge-en-icl (zero-shot) | bge-en-icl (few-shot) |
|---|---|---|---|---|---|---|---|
| ArguAna | 68.21 | 77.37 | 64.27 | 62.34 | 65.27 | 82.76 | 83.08 |
| ClimateFEVER | 34.72 | 39.37 | 45.88 | 34.43 | 46.11 | 45.35 | 45.43 |
| CQADupStack | 50.51 | 47.94 | 46.43 | 46.11 | 47.75 | 47.23 | 47.31 |
| DBPEDIA | 48.29 | 51.37 | 52.42 | 51.21 | 52.28 | 50.42 | 51.63 |
| FEVER | 87.77 | 90.38 | 95.11 | 92.16 | 94.83 | 91.96 | 92.83 |
| FiQA2018 | 63.10 | 60.04 | 62.03 | 61.77 | 60.48 | 58.77 | 59.67 |
| HotpotQA | 79.92 | 83.26 | 73.08 | 81.36 | 76.67 | 84.98 | 85.14 |
| MSMARCO | 46.49 | 45.71 | 45.98 | 42.18 | 45.22 | 46.72 | 46.79 |
| NFCorpus | 38.04 | 38.11 | 40.60 | 41.34 | 42.00 | 40.69 | 41.85 |
| Natural Question | 71.22 | 71.45 | 67.00 | 73.96 | 71.80 | 73.85 | 73.88 |
| QuoraRetrieval | 89.21 | 90.04 | 90.09 | 89.58 | 90.03 | 91.02 | 90.95 |
| SCIDOCS | 20.19 | 26.93 | 28.91 | 24.87 | 26.64 | 25.25 | 25.26 |
| SciFact | 78.43 | 72.05 | 79.06 | 85.91 | 80.09 | 78.33 | 79.09 |
| Touche2020 | 28.38 | 30.26 | 30.57 | 28.18 | 29.94 | 29.67 | 30.48 |
| TREC-COVID | 85.88 | 64.27 | 82.26 | 87.28 | 85.98 | 78.11 | 79.08 |
| BIOSSES | 85.59 | 85.74 | 81.37 | 87.60 | 83.11 | 86.35 | 86.47 |
| SICK-R | 82.80 | 82.66 | 79.28 | 77.01 | 82.89 | 83.87 | 83.87 |
| STS12 | 76.22 | 77.71 | 79.55 | 75.67 | 80.09 | 77.73 | 78.14 |
| STS13 | 86.30 | 87.45 | 88.83 | 82.40 | 89.68 | 85.98 | 86.59 |
| STS14 | 82.09 | 83.48 | 83.87 | 79.93 | 85.07 | 82.34 | 82.83 |
| STS15 | 87.24 | 87.63 | 88.54 | 85.82 | 89.39 | 87.35 | 87.77 |
| STS16 | 84.77 | 86.70 | 86.49 | 84.50 | 87.15 | 86.54 | 87.04 |
| STS17 | 87.42 | 91.18 | 88.73 | 88.93 | 91.35 | 91.25 | 91.25 |
| STS22 | 69.85 | 69.02 | 66.88 | 67.10 | 68.10 | 68.08 | 70.07 |
| STSBenchmark | 86.14 | 87.25 | 86.85 | 83.60 | 88.23 | 87.92 | 88.42 |
| SummEval | 31.20 | 31.20 | 31.35 | 30.71 | 31.49 | 30.75 | 30.77 |
| SprintDuplicateQuestions | 95.94 | 90.94 | 92.82 | 97.62 | 96.04 | 95.06 | 97.23 |
| TwitterSemEval2015 | 78.73 | 79.64 | 77.96 | 78.57 | 80.58 | 78.54 | 79.34 |
| TwitterURLCorpus | 86.05 | 86.95 | 86.59 | 88.03 | 87.58 | 87.19 | 87.84 |
| AmazonCounterfactual | 95.12 | 89.48 | 91.31 | 92.72 | 92.87 | 92.88 | 93.15 |
| AmazonPolarity | 97.14 | 96.90 | 97.50 | 97.31 | 97.16 | 96.86 | 96.98 |
| AmazonReviews | 55.47 | 61.60 | 62.56 | 61.04 | 59.36 | 61.28 | 61.46 |
| Banking77 | 90.34 | 92.53 | 87.57 | 90.02 | 89.79 | 91.42 | 91.49 |
| Emotion | 91.71 | 92.97 | 79.45 | 93.37 | 84.29 | 93.31 | 93.36 |
| Imdb | 97.06 | 96.66 | 96.75 | 96.80 | 96.66 | 96.91 | 96.91 |
| MassiveIntent | 80.07 | 82.05 | 85.41 | 85.97 | 85.83 | 82.26 | 82.93 |
| MassiveScenario | 81.74 | 84.40 | 89.77 | 90.61 | 90.20 | 83.92 | 85.60 |
| MTOPDomain | 96.51 | 98.61 | 99.04 | 98.58 | 99.01 | 97.99 | 98.42 |
| MTOPIntent | 89.77 | 95.51 | 91.88 | 91.30 | 92.78 | 93.56 | 94.00 |
| ToxicConversations | 92.60 | 87.34 | 85.12 | 91.14 | 88.76 | 93.16 | 93.17 |
| TweetSentimentExtraction | 80.60 | 78.86 | 72.58 | 79.70 | 74.84 | 79.90 | 79.93 |
| Arxiv-P2P | 53.76 | 54.91 | 54.46 | 54.02 | 55.44 | 54.42 | 54.44 |
| Arxiv-S2S | 49.59 | 50.28 | 51.74 | 48.82 | 50.66 | 49.17 | 49.33 |
| Biorxiv-P2P | 48.15 | 52.64 | 50.09 | 50.76 | 50.68 | 52.32 | 53.05 |
| Biorxiv-S2S | 44.74 | 49.20 | 46.65 | 46.57 | 46.87 | 48.38 | 48.38 |
| Medrxiv-P2P | 39.24 | 45.81 | 46.23 | 46.66 | 46.87 | 46.13 | 45.86 |
| Medrxiv-S2S | 36.98 | 44.11 | 44.13 | 44.18 | 44.65 | 44.20 | 44.33 |
| Reddit | 63.20 | 56.03 | 73.55 | 62.92 | 72.86 | 71.20 | 72.33 |
| Reddit-P2P | 68.01 | 65.83 | 74.13 | 72.74 | 75.27 | 72.17 | 72.72 |
| StackExchange | 74.99 | 66.21 | 79.86 | 76.48 | 80.29 | 81.29 | 81.32 |
| StackExchange-P2P | 42.04 | 45.74 | 49.41 | 48.29 | 49.57 | 45.53 | 46.05 |
| TwentyNewsgroups | 60.13 | 70.44 | 53.91 | 66.42 | 61.43 | 68.51 | 68.98 |
| AskUbuntuDupQuestions | 67.50 | 64.59 | 67.58 | 66.71 | 67.33 | 64.80 | 65.15 |
| MindSmallRerank | 30.82 | 31.79 | 33.36 | 31.26 | 33.05 | 30.60 | 30.60 |
| SciDocsRR | 87.26 | 87.60 | 89.09 | 87.29 | 89.20 | 86.90 | 86.96 |
| StackOverflowDupQuestions | 56.58 | 54.90 | 55.66 | 55.32 | 55.25 | 56.32 | 56.71 |
| **MTEB Average (56)** | 69.32 | 69.88 | 70.24 | 70.31 | 71.19 | 71.24 | **71.67** |

Table 7: MTEB results with Augmented E5-Mistral dataset

.

| Dataset | bge-en-icl (zero-shot) | bge-en-icl (few-shot) |
|---|---|---|
| ArguAna | 55.81 | 55.41 |
| ClimateFEVER | 45.17 | 45.14 |
| CQADupStack | 46.03 | 46.46 |
| DBPEDIA | 50.79 | 51.14 |
| FEVER | 91.96 | 92.42 |
| FiQA2018 | 58.49 | 58.15 |
| HotpotQA | 84.34 | 84.68 |
| MSMARCO | 46.52 | 46.56 |
| NFCorpus | 40.16 | 40.96 |
| Natural Question | 73.56 | 74.01 |
| QuoraRetrieval | 90.79 | 90.89 |
| SCIDOCS | 20.56 | 20.87 |
| SciFact | 78.10 | 79.65 |
| Touche2020 | 33.64 | 34.93 |
| TREC-COVID | 77.89 | 79.95 |
| BIOSSES | 86.80 | 87.49 |
| SICK-R | 83.83 | 83.69 |
| STS12 | 77.80 | 78.39 |
| STS13 | 84.90 | 85.62 |
| STS14 | 82.53 | 82.62 |
| STS15 | 88.33 | 88.52 |
| STS16 | 86.14 | 86.44 |
| STS17 | 91.65 | 91.79 |
| STS22 | 63.79 | 64.83 |
| STSBenchmark | 87.27 | 87.52 |
| SummEval | 29.52 | 30.68 |
| SprintDuplicateQuestions | 94.79 | 96.09 |
| TwitterSemEval2015 | 81.53 | 82.04 |
| TwitterURLCorpus | 87.30 | 87.39 |
| AmazonCounterfactual | 82.40 | 83.36 |
| AmazonPolarity | 88.57 | 92.69 |
| AmazonReviews | 47.25 | 49.85 |
| Banking77 | 87.57 | 88.70 |
| Emotion | 53.74 | 54.24 |
| Imdb | 81.14 | 84.96 |
| MassiveIntent | 77.87 | 79.24 |
| MassiveScenario | 79.77 | 82.00 |
| MTOPDomain | 95.68 | 96.61 |
| MTOPIntent | 85.22 | 88.19 |
| ToxicConversations | 63.58 | 64.68 |
| TweetSentimentExtraction | 63.47 | 63.16 |
| Arxiv-P2P | 47.22 | 48.97 |
| Arxiv-S2S | 42.87 | 45.35 |
| Biorxiv-P2P | 33.17 | 38.37 |
| Biorxiv-S2S | 35.00 | 37.05 |
| Medrxiv-P2P | 28.74 | 30.24 |
| Medrxiv-S2S | 28.10 | 31.45 |
| Reddit | 53.83 | 59.14 |
| Reddit-P2P | 64.40 | 65.51 |
| StackExchange | 57.50 | 68.61 |
| StackExchange-P2P | 34.21 | 36.01 |
| TwentyNewsgroups | 43.65 | 51.40 |
| AskUbuntuDupQuestions | 63.71 | 62.96 |
| MindSmallRerank | 27.90 | 27.90 |
| SciDocsRR | 84.31 | 84.24 |
| StackOverflowDupQuestions | 51.48 | 51.56 |
| **MTEB Average (56)** | 64.67 | 66.08 |

Table 8: MTEB results with E5-Mistral dataset

.

## C  THE NUMBER OF EXAMPLES

| Task | Retr. | Rerank. | Clust. | PairClass. | Class. | STS | Summ. | Avg. |
|---|---|---|---|---|---|---|---|---|
| # of datasets → | 15 | 4 | 11 | 3 | 12 | 10 | 1 | 56 |
| w/ E5-Mistral dataset | | | | | | | | |
| 0-shot examples | 59.59 | 56.85 | 42.61 | 87.87 | 75.52 | 83.30 | 29.52 | 64.67 |
| 1-shot examples | 59.72 | 57.43 | 44.86 | 88.24 | 76.91 | 83.49 | 30.54 | 65.57 |
| 2-shot examples | 59.95 | 56.90 | 45.79 | 88.33 | 77.25 | 83.68 | 30.68 | 65.90 |
| 3-shot examples | 60.10 | 56.94 | 46.31 | 88.51 | 77.59 | 83.66 | 30.71 | 66.12 |
| 4-shot examples | 60.11 | 57.18 | 46.64 | 88.52 | 77.54 | 83.68 | 30.96 | **66.18** |
| 5-shot examples | 60.10 | 57.15 | 46.64 | 88.54 | 77.45 | 83.70 | 30.83 | **66.18** |

Table 9: Results with different number of examples on the MTEB Benchmark.

Previous efforts to apply in-context learning techniques developed for generative models have shown that the number of in-context examples significantly influences their performance. To investigate whether this phenomenon similarly affects embedding models, we conduct a series of experiments varying the number of in-context examples. The results are presented in Table 9. It can be observed that the empirical performance of different tasks shows consistent improvement as the number of examples increases within certain ranges. However, beyond these ranges, the performance stabilizes, with additional examples yielding no further gains. This empirical evidence suggests that five examples are sufficient for most tasks.

## D  THE ORDER OF EXAMPLES

| Task | Retr. | Rerank. | Clust. | PairClass. | Class. | STS | Summ. | Avg. |
|---|---|---|---|---|---|---|---|---|
| # of datasets → | 15 | 4 | 11 | 3 | 12 | 10 | 1 | 56 |
| w/ E5-Mistral dataset | | | | | | | | |
| 3-shot examples (shuffle-1) | 60.10 | 56.94 | 46.31 | 88.51 | 77.59 | 83.66 | 30.71 | 66.12 |
| 3-shot examples (shuffle-2) | 60.16 | 56.99 | 46.23 | 88.46 | 77.65 | 83.67 | 30.58 | 66.13 |
| 3-shot examples (shuffle-3) | 60.14 | 56.96 | 46.18 | 88.52 | 77.59 | 83.60 | 30.80 | 66.10 |
| 3-shot examples (shuffle-4) | 60.13 | 57.07 | 46.29 | 88.54 | 77.65 | 83.61 | 30.64 | **66.14** |

Table 10: Results with different orders of examples on the MTEB Benchmark.

When using in-context learning with generative models, the order in which examples are presented to the model can significantly influence its output. To examine whether the order of examples affects the performance of embedding models, we conduct an experiment involving three examples. We randomly shuffle the order of these examples four times to analyze the potential impact of their ordering on the model's performance, and the results are shown in Table 10. Across the four random shuffles, the overall performance of the model remains relatively stable. This suggests that different orders of the same examples do not significantly impact the final results. The model demonstrates reliable robustness when faced with varying orders of the same examples.

## E  THE SELECTION OF EXAMPLES

In the context of in-context learning for text generation, identical inputs can yield different outputs depending on the examples provided. To determine if selecting examples randomly offers substantial improvements over a zero-shot approach, we compare our default example selection strategy with a random selection approach. In the random selection approach, we perform the selection process three times, labeled as "random selection strategy -1, -2, -3". The results are shown in Table 11. It indicates that both the default and random selection strategies significantly outperform the zero-shot baseline. Therefore, employing a random selection strategy is also a viable method for selecting examples to enhance model performance.

| Task | Retr. | Rerank. | Clust. | PairClass. | Class. | STS | Summ. | Avg. |
|---|---|---|---|---|---|---|---|---|
| # of datasets → | 15 | 4 | 11 | 3 | 12 | 10 | 1 | 56 |
| w/ E5-Mistral dataset | | | | | | | | |
| zero-shot | 59.59 | 56.85 | 42.61 | 87.87 | 75.52 | 83.30 | 29.52 | 64.67 |
| the default selection strategy | 60.08 | 56.67 | 46.55 | 88.51 | 77.39 | 83.69 | 30.68 | 66.08 |
| random selection strategy - 1 | 60.06 | 57.40 | 46.74 | 88.51 | 77.28 | 83.29 | 30.68 | 66.09 |
| random selection strategy - 2 | 60.06 | 57.53 | 46.75 | 88.51 | 76.94 | 83.59 | 30.68 | 66.08 |
| random selection strategy - 3 | 60.04 | 57.44 | 46.68 | 88.51 | 77.36 | 83.50 | 30.68 | **66.13** |

Table 11: Results with different example selection strategies on the MTEB Benchmark.

| Task | Retr. | Rerank. | Clust. | PairClass. | Class. | STS | Summ. | Avg. |
|---|---|---|---|---|---|---|---|---|
| # of datasets → | 15 | 4 | 11 | 3 | 12 | 10 | 1 | 56 |
| w/ E5-Mistral dataset | | | | | | | | |
| rank 8 (zero-shot) | 58.78 | 57.14 | 42.94 | 87.03 | 75.51 | 82.87 | 29.62 | 64.43 |
| rank 8 (few-shot) | 59.36 | 57.26 | 45.85 | 88.36 | 76.90 | 83.21 | 29.82 | 65.60 |
| rank 16 (zero-shot) | 59.55 | 57.32 | 42.95 | 87.58 | 74.99 | 83.16 | 29.77 | 64.62 |
| rank 16 (few-shot) | 59.91 | 57.03 | 46.78 | 88.74 | 76.64 | 83.71 | 30.58 | 65.98 |
| rank 32 (zero-shot) | 59.59 | 56.85 | 42.61 | 87.87 | 75.52 | 83.30 | 29.52 | 64.67 |
| rank 32 (few-shot) | 60.08 | 56.67 | 46.55 | 88.51 | 77.31 | 83.69 | 30.68 | 66.08 |
| rank 64 (zero-shot) | 59.21 | 57.02 | 43.36 | 87.34 | 75.21 | 83.64 | 30.40 | 64.72 |
| rank 64 (few-shot) | 59.83 | 56.83 | 46.78 | 88.54 | 77.51 | 84.08 | 30.39 | **66.18** |
| rank 128 (zero-shot) | 59.35 | 57.24 | 42.69 | 87.68 | 75.59 | 83.44 | 29.67 | 64.70 |
| rank 128 (few-shot) | 59.85 | 57.23 | 46.38 | 88.64 | 77.60 | 83.93 | 30.09 | 66.12 |

Table 12: Results with different lora rank on the MTEB Benchmark.

## F  THE RESULTS OF LORA RANK

We also explore the hyperparameters used for training the model. Specifically, in addition to the LoRA rank of 32 employed in our experiments, we investigate the performance of the model with LoRA ranks of 8, 16, 64, and 128. The experimental results are presented in Table 12. It can be observed that as the LoRA rank increases, the overall performance of the model gradually improves until it stabilizes. However, higher LoRA ranks require more computational resources. Therefore, using a default rank of 32 in our experiments strikes a balance between performance and computational efficiency.

