# OpenReview forum: "Making Text Embedders Few-Shot Learners"
_ICLR.cc/2025/Conference — ICLR 2025 Poster_

### Official Review · Reviewer_zzdF · 2024-10-27

**Soundness:** 4
**Presentation:** 4
**Contribution:** 4
**Rating:** 8
**Confidence:** 4

**Summary:**

This paper introduces in-context learning to enhance text representation. Since decoder-only LLMs show good performance when adapting into text embedding tasks, this paper proposes to inherit LLM's in-context capability to handle unseen tasks effectively.

Unlike previous task-specific prompt engineering, randomly sampling some examples as the demonstration during training enables embedding model to do in-context learning while maintaining the zero-shot capabilities. The paper shows performance improvement when adding task demonstration to encode text representation. This method does not require additional data or model modifications, keeping the original architecture to preserve ICL capabilities.

**Strengths:**

1. This paper proposes a simple and effective method to enhance text representation. Even though ICL is a common way to enhance model's performance in LLM area, applying it into the text representation is quite reasonable. The experiments are strong and show a great advantage to previous methods.

2. This paper makes a comprehensive ablation studies to justify the methods. First, by aligning the experiment setting, the paper shows that few-shot text embedding is better than zero-shot text embedding. Second, the experiment on model architecture comparison shows that keeping the model architecture is crucial for ICL capabilities.

**Weaknesses:**

Unlike zero-shot text representation, ICL is highly sensitive to the chosen demonstrations. There are countless related works discussing that the selection and order of task demonstration affects the final performance. However, in this paper, the demonstration is not discussed well. The paper only mentions "a consistent set of in-context examples is applied to each query".

**Questions:**

1. Following Weaknesses 1, I believe not all of the demonstration can provide a positive influence on text representation. Therefore, I'm curious if you make a demonstration search, or the examples is just randomly sampled, or you report the average score of many demonstrations.

2. I notice that LoRA is activated in training stage. Is it due to the computation budget limitation or performance consideration? If it is only for efficient training, I'd like to know if you have make comparisons on different LoRA rank, and if it is possible to use full training to further improve models 'performance.

---

> ### Author Response · Authors · 2024-11-24
>
> Dear Reviewer zzdF,
>
> Thank you very much for your thorough review and constructive feedback! We greatly appreciate the opportunity to address your questions with the following response.
>
> > **W1**: *Unlike zero-shot text representation, ICL is highly sensitive to the chosen demonstrations. There are countless related works discussing that the selection and order of task demonstration affects the final performance. However, in this paper, the demonstration is not discussed well. The paper only mentions "a consistent set of in-context examples is applied to each query".*
> >
> > **Q1**: *Following Weaknesses 1, I believe not all of the demonstration can provide a positive influence on text representation. Therefore, I'm curious if you make a demonstration search, or the examples is just randomly sampled, or you report the average score of many demonstrations.*
>
> To explore these questions, we conducted the following pilot experiments involving the order and selection method of the in-context examples.
>
> - **The order of examples**. We randomly shuffled the 3-shot examples multiple times and reported their performances on MTEB. According to our results, the order of examples did not significantly impact the overall performance.
>
> |                                  |  **Ret**  | **Rerank** | **Clustering** | **PairClassification** | **Classification** |  **STS**  | **Summ**  |  **Avg**  |
> | :------------------------------: | :-------: | :--------: | :------------: | :--------------------: | :----------------: | :-------: | :-------: | :-------: |
> | **3-shot example (shuffle - 1)** |   60.10   |   56.94    |   **46.31**    |         88.51          |       77.59        |   83.66   |   30.71   |   66.12   |
> | **3-shot example (shuffle - 2)** | **60.16** |   56.99    |     46.23      |         88.46          |     **77.65**      | **83.67** |   30.58   |   66.13   |
> | **3-shot example (shuffle - 3)** |   60.14   |   56.96    |     46.18      |         88.52          |       77.59        |   83.60   | **30.80** |   66.10   |
> | **3-shot example (shuffle - 4)** |   60.13   | **57.07**  |     46.29      |       **88.54**        |     **77.65**      |   83.61   |   30.64   | **66.14** |
>
> - **The selection method**. We explored the following selection methods in the experiment.
>
>   - **Default selection**. In our default implementation, we sampled the k-shot examples from the training set three times and chose the optimal k-shot examples with the highest accuracy on a small subset of the training set.
>   - **Random selection**. We further explored the random selection method, where the k-shot examples were randomly sampled from the training set (we performed random selection three times, denoted as "random selection strategy -1, -2, -3"). The experiment results on MTEB are reported as follows.
>
>   According to the experiment results, both the default selection strategy and the random selection strategy were sufficient to achieve substantial improvements over the zero-shot baseline.
>
>
> |                                    |  **Ret**  | **Rerank** | **Clustering** | PairClassification | **Classification** |  **STS**  | Summ      | **Avg**   |
> | :--------------------------------: | :-------: | :--------: | :------------: | :----------------: | :----------------: | :-------: | --------- | --------- |
> |           **zero-shot**            |   59.59   |   56.85    |     42.61      |       87.87        |       75.52        |   83.30   | 29.52     | 64.67     |
> | **the default selection strategy** | **60.08** |   56.67    |     46.55      |     **88.51**      |     **77.39**      | **83.69** | **30.68** | 66.08     |
> | **random selection strategy - 1**  |   60.06   |   57.40    |     46.74      |     **88.51**      |       77.28        |   83.29   | **30.68** | 66.09     |
> | **random selection strategy - 2**  |   60.06   | **57.53**  |   **46.75**    |     **88.51**      |       76.94        |   83.59   | **30.68** | 66.08     |
> | **random selection strategy - 3**  |   60.04   |   57.44    |     46.68      |     **88.51**      |       77.36        |   83.50   | **30.68** | **66.13** |

---

> > ### Author Response · Authors · 2024-11-24
> >
> > > **Q2**: *I notice that LoRA is activated in training stage. Is it due to the computation budget limitation or performance consideration? If it is only for efficient training, I'd like to know if you have make comparisons on different LoRA rank, and if it is possible to use full training to further improve models 'performance.*
> >
> > According to the previous studies made by RepLLaMA [1] and other LLM-based embedders, the direct full-parameter fine-tuning of LLM-based embedding models is prone to overfitting, leading to much higher costs and suboptimal performance. As a result, LoRA fine-tuning is regarded as the default training method in this field. Here, we conducted a pilot experiment with LoRA adapters of different ranks (8, 16, 32). According to the results on MTEB, LoRA with rank 32 achieved the highest performance of all.
> >
> > |                         |  **Ret**  | **Rerank** | **Clustering** | **PairClassification** | **Classification** |  **STS**  | **Summ**  |  **Avg**  |
> > | :---------------------: | :-------: | :--------: | :------------: | :--------------------: | :----------------: | :-------: | :-------: | :-------: |
> > | **rank 8 (zero-shot)**  |   58.78   |   57.14    |     42.94      |         87.03          |       75.51        |   82.87   |   29.62   |   64.43   |
> > |  **rank 8 (few-shot)**  |   59.36   | **57.26**  |     45.85      |         88.36          |       76.90        |   83.21   |   29.82   |   65.60   |
> > | **rank 16 (zero-shot)** |   59.55   |   57.23    |     42.95      |         87.58          |       74.99        |   83.16   |   29.77   |   64.62   |
> > | **rank 16 (few-shot)**  |   59.91   |   57.03    |   **46.78**    |       **88.74**        |       76.64        | **83.71** |   30.58   |   65.98   |
> > | **rank 32 (zero-shot)** |   59.59   |   56.85    |     42.61      |         87.87          |       75.52        |   83.30   |   29.52   |   64.67   |
> > | **rank 32 (few-shot)**  | **60.08** |   56.67    |     46.55      |         88.51          |     **77.31**      |   83.69   | **30.68** | **66.08** |
> >
> > [1] Fine-tuning llama for multi-stage text retrieval, Ma et al.

---

> ### Comment · Reviewer_zzdF · 2024-11-24
>
> Thanks for your response! Now I see that the order and the selection of few-shot examples does not make a big influence on the performance. I'd like to increase my score to 8. It is indeed a simple but effective method.
>
> Nevertheless, I still want to discuss the LoRA part. Even though there are references showing that full-rank tuning is prune to over-fitting, but your experiments show that larger LoRA rank makes a better result. If you can show that the bigger rank (e.g. 64, 128) will make the model overfitting, it will be much more persuasive.

---

> > ### Author Response · Authors · 2024-11-25
> >
> > Thanks a lot for your acknowledgement!  We will investigate whether further increasing LoRA rank can bring extra improvements.  New results will be posted once the experiment completes.

---

> > > ### Author Response · Authors · 2024-11-27
> > >
> > > Dear reviewer,
> > >
> > > We have completed the additional experiments for LoRA rank-64 and LoRA rank-128, whose results are shown in the following table. As we can see, rank-64 still slightly improves upon rank-32, while rank-128 results in a similar performance as rank-64. It suggests that rank-64 is the optimal choice for the current setting of our problem.
> > >
> > > |                          |  **Ret**  | **Rerank** | **Clustering** | **PairClassification** | **Classification** |  **STS**  | **Summ**  |  **Avg**  |
> > > | :----------------------: | :-------: | :--------: | :------------: | :--------------------: | :----------------: | :-------: | :-------: | :-------: |
> > > |  **rank 8 (zero-shot)**  |   58.78   |   57.14    |     42.94      |         87.03          |       75.51        |   82.87   |   29.62   |   64.43   |
> > > |  **rank 8 (few-shot)**   |   59.36   | **57.26**  |     45.85      |         88.36          |       76.90        |   83.21   |   29.82   |   65.60   |
> > > | **rank 16 (zero-shot)**  |   59.55   |   57.23    |     42.95      |         87.58          |       74.99        |   83.16   |   29.77   |   64.62   |
> > > |  **rank 16 (few-shot)**  |   59.91   |   57.03    |   **46.78**    |       **88.74**        |       76.64        |   83.71   |   30.58   |   65.98   |
> > > | **rank 32 (zero-shot)**  |   59.59   |   56.85    |     42.61      |         87.87          |       75.52        |   83.30   |   29.52   |   64.67   |
> > > |  **rank 32 (few-shot)**  | **60.08** |   56.67    |     46.55      |         88.51          |       77.31        |   83.69   | **30.68** |   66.08   |
> > > | **rank 64 (zero-shot)**  |   59.21   |   57.02    |     43.36      |         87.34          |       75.21        |   83.64   |   30.40   |   64.72   |
> > > |  **rank 64 (few-shot)**  |   59.83   |   56.83    |     46.78      |         88.54          |       77.51        | **84.08** |   30.39   | **66.18** |
> > > | **rank 128 (zero-shot)** |   59.35   |   57.24    |     42.69      |         87.68          |       75.59        |   83.44   |   29.67   |   64.70   |
> > > | **rank 128 (few-shot)**  |   59.85   |   57.23    |     46.38      |         88.64          |     **77.60**      |   83.93   |   30.09   |   66.12   |

---

### Official Review · Reviewer_uYsr · 2024-10-27

**Soundness:** 3
**Presentation:** 3
**Contribution:** 3
**Rating:** 6
**Confidence:** 4

**Summary:**

The paper proposes an LLM-based text embedding model that supports in-context learning. The training recipe is fairly simple and straightforward and yields impressive empirical results in comparable settings. The experiments also validate the modeling choices, and suggest that complex architectural changes are not required for performance gains.

**Strengths:**

- Simplicity and intuitiveness of the technique.
- Strong and extensive empirical results
- Thorough ablations

I especially liked the ablation from Section 4.4, which shows that the simple ICL setup outperforms other architectural choices. Such ablations are necessary for disentangling the performance gains due to data and architectural changes.

**Weaknesses:**

**The submission is not following the ICLR template because line numbers are missing**

My main concern regarding the paper is that some key implementation details are missing (See Questions). Given the extra page limit and verbose Section 3.1 and Figure 2, I think the authors could have done a better job in covering those details.

Other than that, NV-Embed2 results can also be included in the revision.

Other comments about writing:
- The right two subfigures in Figure 2 are not really required. I would suggest removing them altogether or moving the figures to the appendix.
- The use of "full data" vs. "public data" feels like a misrepresentation of "full data," given that "full data" is also public. I suggest using different terms.

Typos:
- Use \citet{}. Section 2, second paragraph, second line
- Section 3.1 - "task_definition" -> Fix the opening quote.
- Section 4.4, third paragraph - "align" -> "aligned"
- Training Details: "conduct the process over a single epoch" - I have never seen such language for describing "trained for a single epoch".

**Questions:**

- How exactly are the query and passage representations obtained? The last line of Section 3.1 suggests that the two are obtained using <EOS> embeddings. Are there two <EOS> tokens?
- Why are passages/documents truncated to 256 tokens, given that Mistral has 32K tokens and there are only a maximum of 5 documents? Are the gains on Long Doc evals in line with other gains, given that nothing in the architecture/training should benefit long documents in particular?
- In Evaluation, it's said that "a consistent set" is used for ICL evals. What is a consistent set? Do you mean fixed? How many examples?
- In Section 4.4, are all the ablations models trained similarly to the baseline model?

---

> ### Author Response · Authors · 2024-11-24
>
> Dear Reviewer uYsr,
>
> Thank you very much for your thorough review and constructive feedback! We greatly appreciate the opportunity to address your questions with the following response.
>
>
> > **W1**: *My main concern regarding the paper is that some key implementation details are missing (See Questions). Given the extra page limit and verbose Section 3.1 and Figure 2, I think the authors could have done a better job in covering those details.*
> >
> > **W2**: *Other than that, NV-Embed2 results can also be included in the revision.*
> >
> > **W3**: *Other comments about writing.*
>
> - Thanks a lot for pointing out the issues about writing and format. We have revised the paper accordingly based on these suggestions.
> - We've also added NV-Embed-v2 to the revised paper (line 337, 346, 385).
>
>
> > **Q1**: *How exactly are the query and passage representations obtained? The last line of Section 3.1 suggests that the two are obtained using \<EOS\> embeddings. Are there two \<EOS\> tokens?*
>
> We append an \<EOS\> token to the end of the input when encoding the query and passage. The final hidden state of the \<EOS\> token is used as the embedding. We apply the same \<EOS\> token for encoding both the query and the passage.
>
> > **Q2**: *Why are passages/documents truncated to 256 tokens, given that Mistral has 32K tokens and there are only a maximum of 5 documents? Are the gains on Long Doc evals in line with other gains, given that nothing in the architecture/training should benefit long documents in particular?*
>
> - For the majority of our training tasks, 256 tokens are sufficient to cover a complete passage/document in the datasets.
> - Excessively long inputs will severely restrict the batch size, which is unfavorable to the training quality.
> - During evaluation, the examples are introduced without truncation. Despite that the training is conducted with shorter lengths, our method remains effective when it is directly applied to such longer inputs (as shown in Table 3, lines 378-390).
>
>
> > **Q3**: *In Evaluation, it's said that "a consistent set" is used for ICL evals. What is a consistent set? Do you mean fixed? How many examples?*
>
> - We use a fixed set of examples for all queries within the same dataset.
> - The number of examples is randomly determined for each dataset in the original paper.
> - We further include additional experiments by fixing the number of examples (using E5-Mistral data).
>
> |        MTEB        |  **Ret**  | **Rerank** | **Clustering** | **PairClassification** | **Classification** |  **STS**  | **Summ**  |  **Avg**  |
> | :----------------: | :-------: | :--------: | :------------: | :--------------------: | :----------------: | :-------: | :-------: | :-------: |
> | **0-shot example** |   59.59   |   56.85    |     42.61      |         87.87          |       75.52        |   83.30   |   29.52   |   64.67   |
> | **1-shot example** |   59.72   | **57.43**  |     44.86      |         88.24          |       76.91        |   83.49   |   30.54   |   65.57   |
> | **2-shot example** |   59.98   |   56.90    |     45.79      |         88.33          |       77.25        |   83.68   |   30.68   |   65.90   |
> | **3-shot example** |   60.10   |   56.94    |     46.31      |         88.51          |     **77.59**      |   83.66   |   30.71   |   66.12   |
> | **4-shot example** | **60.11** |   57.18    |   **46.64**    |         88.52          |       77.44        |   83.68   | **30.96** | **66.18** |
> | **5-shot example** |   60.10   |   57.15    |   **46.64**    |       **88.54**        |       77.45        | **83.70** |   30.83   | **66.18** |
>
> > **Q4**: *In Section 4.4, are all the ablations models trained similarly to the baseline model?*
>
> Yes, all the ablations models are trained using the same settings as the baseline model.

---

> ### Comment · Reviewer_uYsr · 2024-11-24
>
> Thanks for answering all my questions in detail and for sharing the additional results.
>
> I wanted to follow up on a few things from the response.
>
> > The number of examples is randomly determined for each dataset in the original paper.
>
> What do you mean by randomly determined?
>
> > We further include additional experiments by fixing the number of examples (using E5-Mistral data).
>
> The result suggests an almost monotonic performance between the number of examples and performance which is quite encouraging.

---

> ### Author Response · Authors · 2024-11-25
>
> Thanks a lot for your reply! Here are our answers to your following-up questions.
>
> > What do you mean by randomly determined?
>
> We randomly sample the value of k between 1~8 for each dataset. Then, we further sample the k-shot examples as our demonstration for this dataset.
>
> > The result suggests an almost monotonic performance between the number of examples and performance which is quite encouraging.
>
> Based on our experiment result, we can have the following observations:
> - Within certain ranges, the empirical performance of different tasks improves consistently as the number of examples increases.
> - Beyond these ranges, the performance becomes stabilized where additional examples do not lead to further gains.
> - It empirically suggests that 5 examples are sufficient for most of the tasks .

---

> > ### Author Response · Authors · 2024-11-27
> >
> > Dear Reviewer uYsr,
> >
> > We've also included our clarifications and the extra baseline to the newly updated submission. Please feel free to let us know whether there are any further questions about these issues.
> >
> > Thanks, \
> > The authors

---

### Official Review · Reviewer_mLW8 · 2024-10-31

**Soundness:** 3
**Presentation:** 3
**Contribution:** 2
**Rating:** 6
**Confidence:** 4

**Summary:**

This paper proposes to include in-context examples for better text embedding. They do so by finetuning for an epoch in a traditional contrastive learning/embedding-training setup but with in-context examples included in the training data.

**Strengths:**

- This method does appear to slightly improve the performance of text embedding models even at the 7B scale.
- Novelty: In-context learning has been shown to be effective for language models in many scenarios; as far as I'm aware, this is the first work to explore in-context learning at the 7B scale.
- The paper makes a number of other empirical contributions, analyzing factors such as bidirectional attention and pooling as well as details of whether instructions should be added to passages or queries or both.

**Weaknesses:**

- In context examples have been shown to be most useful when a language model needs to learn a template or format for doing a task; in many cases, this is their *only* contribution (as opposed to actually teaching semantic information about the task at hand). This is not useful for embedding models because embedding models always have the same task format (outputting an embedding).
- A major weakness is that this obviously makes the embedding process slower, and it's not clear by quite how much or what the tradeoff is. The performance gains are quite marginal (71.2 -> 71.7 on MTEB) and practitioners would need to balance this with the
- Similarly, it's not clear how much of the performance comes from increasing the number of tokens (and consequently FLOPs) that the model can use at test-time vs. actually incorporating new information into the model. A useful ablation would be training a model with the same number of tokens as a typical ICL example, but with a single token substituted for all ICL tokens (a-la the "let's think dot-by-dot" work). I would suspect that most of the performance comes simply from increased computation at test time.
- Little analysis is shown: beyond questions at the systems level, another clear demonstration of utility would come from comparing model performance to number of shots in context. If increasing the number of in-context examples also increases performance, that would provide a compelling case for practitioners.
- Finally, does this help more out-of-domain? One would expect in-context examples to be most useful (or perhaps even *only* useful) when the test task is most different from the training task. Is this true in your case?

**Questions:**

- Does performance get bewtter by using more examples in context?
- At test time, are few-shot examples selected per-example or per-dataset?
- Do you pool over instruction tokens?
- Do you pool over ICL tokens?

---

> ### Author Response · Authors · 2024-11-24
>
> Dear Reviewer mLW8,
>
> Thank you very much for your thorough review and constructive feedback! We greatly appreciate the opportunity to address your questions with the following response.
>
> > **W1**: *In context examples have been shown to be most useful when a language model needs to learn a template or format for doing a task; in many cases, this is their only contribution (as opposed to actually teaching semantic information about the task at hand). This is not useful for embedding models because embedding models always have the same task format (outputting an embedding).*
>
> - For our problem, the in-context examples are introduced to adapt the models for different embedding tasks. For example, by giving "*example question*: *example answer*", the model is adapted for QA retrieval tasks; by giving "*example document*: *example label*", the model is adapted for classification tasks.
> - In other words, the in-context examples are used to prompt the model to specialize the embedding for the user's interested task. This approach aligns conceptually with the principles of prompt tuning, which applies for both generation and representation tasks [1, 2, 3].
> - The impact of adaptation from in-context examples is demonstrated by our experimental studies. Please continue to check our detailed analysis in the following discussion.
>
> [1] The power of scale for parameter-efficient prompt tuning, Lester et al.
> [2] P-Tuning: Prompt Tuning Can Be Comparable to Fine-tuning Across Scales and Tasks, Liu et al.
> [3] Making Pre-trained Language Models Better Few-shot Learners, Gao et al.
>
> > **W2**: *A major weakness is that this obviously makes the embedding process slower, and it's not clear by quite how much or what the tradeoff is. The performance gains are quite marginal (71.2 -> 71.7 on MTEB) and practitioners would need to balance this with the*
> >
> > **W5**: *Finally, does this help more out-of-domain? One would expect in-context examples to be most useful (or perhaps even only useful) when the test task is most different from the training task. Is this true in your case?*
>
> - In-context examples are particularly useful to adapt the embedding models for unseen tasks (i.e., out-of-domain scenarios), which aligns with your assumption.
> - This point is illustrated by the experiment with the E5-Mistral data (64.67 -> 66.08), where most of the MTEB tasks are held out of the training data.
>
> |                       MTEB                        |  **Ret**  | **Rerank** | **Clustering** | **PairClassification** | **Classification** |  **STS**  | **Summ**  |  **Avg**  |
> | :-----------------------------------------------: | :-------: | :--------: | :------------: | :--------------------: | :----------------: | :-------: | :-------: | :-------: |
> | **icl-embedder (E5-Mistral dataset) (zero-shot)** |   59.59   | **56.85**  |     42.61      |         87.87          |       75.52        |   83.30   |   29.52   |   64.67   |
> | **icl-embedder (E5-Mistral dataset) (few-shot)**  | **60.08** |   56.67    |   **46.55**    |       **88.51**        |     **77.31**      | **83.69** | **30.68** | **66.08** |
>
> - This point is further emphasized by AIR-Bench, where unseen tasks are introduced for the embedding models. Specifically, it demonstrates improvements from 53.60 to 55.92 on top of the few-shot examples.
>
> |                  AIR-Bench (QA)                   | **wiki**  |  **web**  | **news**  | **healthcare** |  **law**  | **finance** | **arxiv** |  msmarco  |  **Avg**  |
> | :-----------------------------------------------: | :-------: | :-------: | :-------: | :------------: | :-------: | :---------: | :-------: | :-------: | :-------: |
> | **icl-embedder (E5-Mistral dataset) (zero-shot)** |   64.82   |   54.96   |   55.82   |     57.06      |   28.87   |    54.46    |   49.60   |   63.25   |   53.60   |
> | **icl-embedder (E5-Mistral dataset) (few-shot)**  | **66.68** | **56.38** | **57.17** |   **59.54**    | **32.03** |  **58.81**  | **51.36** | **65.05** | **55.92** |
>
> - Although it remains effective for in-domain applications, its improvement effect is mitigated due to in-domain fine-tuning (71.2 -> 71.7).
> - Although in-context examples introduce additional encoding costs, they only affect query processing. Given their significant impact on out-of-domain applications, this extra cost can still be affordable in practice. (Besides, knowing that our in-context examples are fixed for each task, the encoding result can be cached and shared by all queries. Therefore, the extra cost can be significantly reduced even further.)

---

> ### Author Response · Authors · 2024-11-24
>
> > **W3**: *Similarly, it's not clear how much of the performance comes from increasing the number of tokens (and consequently FLOPs) that the model can use at test-time vs. actually incorporating new information into the model. A useful ablation would be training a model with the same number of tokens as a typical ICL example, but with a single token substituted for all ICL tokens (a-la the "let's think dot-by-dot" work). I would suspect that most of the performance comes simply from increased computation at test time.*
>
> To explore the impact of simply increasing the number of tokens, we conducted a pilot experiment in which we replaced the in-context examples with an equivalent number of periods ("."), and re-trained the embedding model using E5-Mistral data. We got the following result from our pilot experiment:
>
> |                                                   |  **Ret**  | **Rerank** | **Clustering** | **PairClassification** | **Classification** |  **STS**  | **Summ**  |  **Avg**  |
> | :-----------------------------------------------: | :-------: | :--------: | :------------: | :--------------------: | :----------------: | :-------: | :-------: | :-------: |
> | **icl-embedder (E5-Mistral dataset) (zero-shot)** |   59.59   | **56.85**  |     42.61      |         87.87          |       75.52        |   83.30   |   29.52   |   64.67   |
> | **icl-embedder (E5-Mistral dataset) (few-shot)**  | **60.08** |   56.67    |   **46.55**    |       **88.51**        |     **77.31**      | **83.69** | **30.68** | **66.08** |
> |  **substituttion with "." (E5-Mistral dataset)**  |   58.87   |   55.91    |     39.40      |         84.51          |       72.63        |   80.09   |   30.24   |   62.43   |
>
> As we can observe, the substitution does not improve the performance of the ICL-embedder (zero-shot). Instead, the performance declines from 64.67 to 62.43 due to the introduction of unrelated information.
>
> > **W4**: *Little analysis is shown: beyond questions at the systems level, another clear demonstration of utility would come from comparing model performance to number of shots in context. If increasing the number of in-context examples also increases performance, that would provide a compelling case for practitioners.*
> >
> > **Q1**: *Does performance get better by using more examples in context?*
>
> - Following this suggestion, we conducted an extra experiment where the number of ICL examples is increased from 0 to 5. The experiment results are reported as follows:
>
> |        MTEB        |  **Ret**  | **Rerank** | **Clustering** | **PairClassification** | **Classification** |  **STS**  | **Summ**  |  **Avg**  |
> | :----------------: | :-------: | :--------: | :------------: | :--------------------: | :----------------: | :-------: | :-------: | :-------: |
> | **0-shot example** |   59.59   |   56.85    |     42.61      |         87.87          |       75.52        |   83.30   |   29.52   |   64.67   |
> | **1-shot example** |   59.72   | **57.43**  |     44.86      |         88.24          |       76.91        |   83.49   |   30.54   |   65.57   |
> | **2-shot example** |   59.98   |   56.90    |     45.79      |         88.33          |       77.25        |   83.68   |   30.68   |   65.90   |
> | **3-shot example** |   60.10   |   56.94    |     46.31      |         88.51          |     **77.59**      |   83.66   |   30.71   |   66.12   |
> | **4-shot example** | **60.11** |   57.18    |   **46.64**    |         88.52          |       77.44        |   83.68   | **30.96** | **66.18** |
> | **5-shot example** |   60.10   |   57.15    |   **46.64**    |       **88.54**        |       77.45        | **83.70** |   30.83   | **66.18** |
>
> Based on the above results, the following observations can be made:
>
> - Within certain ranges, the empirical performance of different tasks improves consistently as the number of examples increases.
> - Beyond these ranges, the performance becomes stabilized where additional examples do not lead to further gains.
> - It empirically suggests that 5 examples are sufficient for most of the tasks .
>
> > **Q2**: *At test time, are few-shot examples selected per-example or per-dataset?*
>
> We use a fixed set of examples for all queries within the same dataset.
>
> > **Q3**: *Do you pool over instruction tokens?*
> >
> > **Q4**: *Do you pool over ICL tokens?*
>
> Following the setting in GRITLM [4], the mean pooling only involves the query's tokens, where the instruction's tokens and the examples' tokens are omitted from this operation.
>
> [4]  Generative representational instruction tuning, Muennighoff et al.

---

> > ### Comment · Reviewer_mLW8 · 2024-11-26
> >
> > Thanks for the thoughtful reply. I don't think my main concerns were addressed, such as whether ICL examples are truly useful for out-of-domain tasks or just doing some variant of multitask learning. I will keep my score at a 6.

---

> ### Author Response · Authors · 2024-11-28
>
> Dear Reviewer mLW8,
>
> We highly appreciate your valuable feedback to us! In case our previous response is too wordy to comprehend, we would like to highlight its key points as follows.
>
> - Our method achieves a much more significant improvement in the **out-of-domain evaluations**: 64.67->66.08 on MTEB and 53.60->55.92 on AIR-Bench (the detailed result is presented in our previous response). In fact, the training data of E5-Mistral is mainly made up of retrieval task, where other tasks, like classification, clustering, pairwise, are not included in fine-tuning. Such a result indicates that the improvement is resulted from the embedding model's in-context learning capability, rather than multi-task learning.
>  - The extra latency is moderate given that 1) it only influences **query processing**, 2) the in-context examples are fixed for each task, thus can be significantly **accelerated by KV cache**. To demonstrate the acceleration effect of KV cache, we perform the following experiment, where the query encoding time can be reduced by more than 4 times in 3-shot setting (35.75->8.07). While there is an additional time cost compared to the zero-shot setting, it is acceptable in many scenarios, allowing users to trade off embedding quality and inference latency effectively (i.e., performance improvement through scaling-up of runtime computation, which is in the same spirit of GPT-o1).
>
> |         Method          | Encoding Latency (ms) |
> | :-----------------------: | :----------------: |
> |     without examples      |      4.26      |
> | 1-shot examples (w/ KV cache)  |        5.56        |
> | 1-shot examples (w/o KV cache) |       14.51        |
> | 2-shot examples (w/ KV cache)  |        6.69        |
> | 2-shot examples (w/o KV cache) |       23.63        |
> | 3-shot examples (w/ KV cache)  |        8.07        |
> | 3-shot examples (w/o KV cache) |       35.75        |
>
> Please feel free to let us know any further questions about these issues. We are looking forward to engaging with the reviewer for more discussions.
>
> Thanks, \
> The authors

---

### Official Review · Reviewer_5eiG · 2024-11-02

**Soundness:** 3
**Presentation:** 3
**Contribution:** 3
**Rating:** 8
**Confidence:** 4

**Summary:**

The paper proposes to train instruction-conditional embedding models to take few-shot examples as inputs. Previous work trained embedding models to take instructions as input by contrastive-loss training on a collection of classification / question-answering datasets. This paper’s approach is similar and differs only in that in addition to the instruction during the training one adds several positive examples to the prompt. This additional conditioning leads to significant gains on AIR-Bench and MTEB benchmarks.

**Strengths:**

This is a very straight-forward paper, it is easy to read, it presents compelling, though not jaw-dropping results.

**Weaknesses:**

- It would be nice to see an experiment on how the number of few-shot examples impacts performance.
- The discussion about overlap between training data and MTEB was a bit difficult to follow.

**Questions:**

The "more comprehensive" dataset that you train your model on still consists of public datasets, doesn't it? I find the way you chose to name these two different experimental settings a bit misleading.

---

> ### Author Response · Authors · 2024-11-24
>
> Dear Reviewer 5eiG,
>
> Thank you very much for your thorough review and constructive feedback! We greatly appreciate the opportunity to address your questions with the following response.
>
> > **W1**: *It would be nice to see an experiment on how the number of few-shot examples impacts performance.*
>
> Following the above suggestion, we investigated the effect of the number of examples (denoted as "k-shot" examples) on performance. The experimental results on the MTEB dataset are summarized below.
>
> |        MTEB        |  **Ret**  | **Rerank** | **Clustering** | **PairClassification** | **Classification** |  **STS**  | **Summ**  |  **Avg**  |
> | :----------------: | :-------: | :--------: | :------------: | :--------------------: | :----------------: | :-------: | :-------: | :-------: |
> | **0-shot example** |   59.59   |   56.85    |     42.61      |         87.87          |       75.52        |   83.30   |   29.52   |   64.67   |
> | **1-shot example** |   59.72   | **57.43**  |     44.86      |         88.24          |       76.91        |   83.49   |   30.54   |   65.57   |
> | **2-shot example** |   59.98   |   56.90    |     45.79      |         88.33          |       77.25        |   83.68   |   30.68   |   65.90   |
> | **3-shot example** |   60.10   |   56.94    |     46.31      |         88.51          |     **77.59**      |   83.66   |   30.71   |   66.12   |
> | **4-shot example** | **60.11** |   57.18    |   **46.64**    |         88.52          |       77.44        |   83.68   | **30.96** | **66.18** |
> | **5-shot example** |   60.10   |   57.15    |   **46.64**    |       **88.54**        |       77.45        | **83.70** |   30.83   | **66.18** |
>
> Based on these results, the following observations can be made:
>
> - Within certain ranges, the empirical performance of different tasks improves consistently as the number of examples increases.
> - Beyond these ranges, the performance becomes stabilized where additional examples do not lead to further gains.
> - It empirically suggests that 5 examples are sufficient for most of the tasks.
>
> > **W2**: *The discussion about overlap between training data and MTEB was a bit difficult to follow.*
> >
> > **Q1**: *The "more comprehensive" dataset that you train your model on still consists of public datasets, doesn't it? I find the way you chose to name these two different experimental settings a bit misleading.*
>
> - To avoid ambiguity in the original draft, we've updated the names of the datasets to "**E5-Mistral dataset**" and "**Augmented E5-Mistral dataset**".
> - The E5-Mistral dataset is the one originally used by E5-Mistral [1]. It primarily consists of some public data and MTEB in-domain datasets for the retrieval task. In contrast, the Augmented E5-Mistral dataset is a new dataset introduced in our work, which includes additional in-domain datasets for other MTEB tasks, such as STS and classification. Detailed specifications for both datasets are provided in lines 270–302 of the revised paper.
> - In-context examples enhance the empirical performance on both datasets. However, the improvement is more noticeable on the E5-Mistral dataset, indicating that in-context learning is particularly effective in zero-shot applications.
>
> [1] Improving text embeddings with large language models, Wang et al.

---

### Author Response · Authors · 2024-11-24

Dear Reviewers and Chairs,

We sincerely appreciate your thoughtful and constructive feedback! In our response, we have carefully addressed your comments and made necessary updates to the manuscript. We would like to highlight the following results in our response:

- **Implementation Details**: We have included detailed clarifications about the implementation details, regarding the representation of queries and passages, the truncation method, and the example selection strategy.

- **In-Context Learning's Impact**: We have provided a more comprehensive discussion about the effect of in-context learning, which adapts the embedding model for various tasks (e.g., retrieval, classification, clustering, etc.). This adaptation effect is particularly significant for out-of-domain applications.

- **Naming of Datasets**: To avoid ambiguity in the original draft, we have updated the names of the datasets to "E5-Mistral dataset" (used by E5-Mistral) and "Augmented E5-Mistral dataset" (incorporating additional MTEB in-domain datasets).

- **Extra Investigations into the In-Context Examples**: We conducted additional investigations into the impact of the number, order, and selection of examples. Our findings include:
  * Within a certain range, increasing the number of examples improves empirical performance.
  * The order of examples does not affect empirical performance.
  * A simple random selection method is sufficient to achieve competitive empirical performance.

Please refer to our detailed response and the updated manuscript for more information. We look forward to engaging with the reviewers for further questions and discussions.

---

### Meta-Review · Area_Chair_mcXt · 2024-12-22

**Metareview:**

The reviewers agree that this paper makes a valuable contribution to text embedding research by introducing in-context learning capabilities to embedding models. The proposed approach, while conceptually straightforward, effectively leverages LLMs' ICL capabilities and demonstrates consistent improvements across multiple benchmarks including MTEB and AIR-Bench. The experimental evaluation is thorough, showing meaningful improvements over existing methods, particularly in out-of-distribution tasks. While the technical novelty might not be revolutionary, the paper provides useful insights about model architecture choices and presents a practical solution that could benefit the community. The authors' commitment to reproducibility through open-sourced code and models is appreciated. Based on the reviewers' consensus, I recommend accepting this paper.

**Additional Comments On Reviewer Discussion:**

I have read the messages in the discussion period and my opinion has been summarized as in the metareview above. I considered these points in my recommendation.

---

### Decision · Program_Chairs · 2025-01-22

Accept (Poster)